# GPT-Fathom: Benchmarking Large Language Models to Decipher the Evolutionary Path towards GPT-4 and Beyond

## Abstract

With the rapid advancement of large language models (LLMs), there is a pressing need for a comprehensive evaluation suite to assess their capabilities and limitations. Existing LLM leaderboards often reference scores reported in other papers without consistent settings and prompts, which may inadvertently encourage cherry-picking favored settings and prompts for better results. In this work, we introduce GPT-Fathom, an open-source and reproducible LLM evaluation suite built on top of OpenAI Evals[1]. We systematically evaluate 10+ leading LLMs as well as OpenAI's legacy models on 20+ curated benchmarks across 7 capability categories, all under aligned settings. Our retrospective study on OpenAI's earlier models offers valuable insights into the evolutionary path from GPT-3 to GPT-4. Currently, the community is eager to know how GPT-3 progressively improves to GPT-4, including technical details like whether adding code data improves LLM's reasoning capability, which aspects of LLM capability can be improved by SFT and RLHF, how much is the alignment tax, etc. Our analysis sheds light on many of these questions, aiming to improve the transparency of advanced LLMs.

## 1 Introduction

Recently, the advancement of large language models (LLMs) is arguably the most remarkable breakthrough in Artificial Intelligence (AI) in the past few years. Based on the Transformer (Vaswani et al., 2017) architecture, these LLMs are trained on massive Web-scale text corpora. Despite their straightforward method of using a self-supervised objective to predict the next token, leading LLMs demonstrate exceptional capabilities across a range of challenging tasks (Bubeck et al., 2023), even showing a potential path towards Artificial General Intelligence (AGI). With the rapid progress of LLMs, there is a growing demand for better understanding these powerful models, including the distribution of their multi-aspect capabilities, limitations and risks, and directions and priorities of their future improvement. It is critical to establish a carefully curated evaluation suite that measures LLMs in a systematic, transparent and reproducible manner. Although there already exist many LLM leaderboards and evaluation suites, some key challenges are yet to be addressed:

- *Inconsistent settings:* The evaluation settings, such as the number of in-context example "shots", whether Chain-of-Thought (CoT; Wei et al. 2022) prompting is used, methods of answer parsing and metric computation, etc., often differ across the existing LLM works. Moreover, most of the released LLMs do not disclose their prompts used for evaluation, making it difficult to reproduce the reported scores. Different settings and prompts may lead to very different evaluation results, which may easily skew the observations. Yet, many existing LLM leaderboards reference scores from other papers without consistent settings and prompts, which may inadvertently encourage cherry-picking favored settings and prompts for better results. To achieve reliable conclusions, it is crucial to make apples-to-apples LLM comparisons with consistent settings and prompts.

- *Incomplete collection of models and benchmarks:* For the moment, when compared to OpenAI's leading models such as GPT-4, all the other LLMs (particularly open-source models) exhibit a substantial performance gap. In fact, it takes OpenAI nearly three years to evolve from GPT-3 (released in 2020/06) to GPT-4 (released in 2023/03). Existing LLM leaderboards primarily focus on the latest models, while missing a retrospective study on OpenAI's earlier models and its mysterious path from GPT-3 to GPT-4. Besides the coverage of models, many existing works

assess LLMs on merely one or a few aspects of capabilities, which is not sufficient to provide a comprehensive view to deeply understand the strength and weakness of the evaluated LLMs.

- *Insufficient study on model sensitivity:* LLMs are known to be sensitive to the evaluation setting and the formatting of prompt (Liang et al., 2023). However, many existing works only focus on the benchmark score under one specific setting, while overlooking the impacts of model sensitivity on the overall usability of LLMs. In fact, it is unacceptable that a slightly rephrased prompt could cause the LLM to fail in responding it correctly. Due to the lack of systematic study on model sensitivity, this potential vulnerability in LLMs remains not well understood.

These challenges hinder a comprehensive understanding of LLMs. To dispel the mist among LLM evaluations, we introduce GPT-Fathom, an open-source and reproducible LLM evaluation suite developed based on OpenAI Evals[1]. We evaluate 10+ leading open-source and closed-source LLMs on 20+ curated benchmarks in 7 capability categories under aligned settings. We also evaluate legacy models from OpenAI to retrospectively measure their progressive improvement in each capability dimension. Our retrospective study offers valuable insights into OpenAI's evolutionary path from GPT-3 to GPT-4, aiming to help the community better understand this enigmatic path. Our analysis sheds light on many community-concerned questions (e.g., the gap between OpenAI / non-OpenAI models, whether adding code data improves reasoning capability, which aspects of LLM capability can be improved by SFT and RLHF, how much is the alignment tax, etc.). With reproducible evaluations, GPT-Fathom serves as a standard gauge to pinpoint the position of emerging LLMs, aiming to help the community measure and bridge the gap with leading LLMs. We also explore the impacts of model sensitivity on evaluation results with extensive experiments of various settings.

Benchmarks constantly play a pivotal role in steering the evolution of AI and, of course, directing the advancement of LLMs as well. There are many great existing LLM evaluation suites. By comparing GPT-Fathom with previous works, we summarize the major difference as follows: 1) HELM (Liang et al., 2023) primarily uses answer-only prompting (without CoT) and has not included the latest leading models such as GPT-4 (as of the time of writing); 2) Open LLM Leaderboard (Beeching et al., 2023) focuses on open-source LLMs, while we jointly consider leading closed-source and open-source LLMs; 3) OpenCompass (Contributors, 2023) evaluates latest open-source and closed-source LLMs (all released after 2023/03), while we cover both leading LLMs and OpenAI's earlier models to decipher the evolutionary path from GPT-3 to GPT-4; 4) InstructEval (Chia et al., 2023) is designed for evaluating instruction-tuned LLMs, while we evaluate both base and SFT / RLHF models; 5) AlpacaEval (Li et al., 2023) evaluates on simple instruction-following tasks as a quick and cheap proxy of human evaluation, while we provide systematic evaluation of various aspects of LLM capabilities; 6) Chatbot Arena (Zheng et al., 2023) evaluates human user's dialog preference with a Elo rating system, while we focus on automatic and reproducible evaluation over popular benchmarks; 7) Chain-of-Thought Hub (Fu et al., 2023) focuses on evaluating the reasoning capability of LLMs with CoT prompting, while we support both CoT and answer-only prompting settings and evaluate various aspects of LLM capabilities.

The key contributions of our work are summarized as follows:

- *Systematic and reproducible evaluations under aligned settings:* We provide accurate evaluations of 10+ leading LLMs on 20+ curated benchmarks across 7 capability categories. We carefully align the evaluation setting for each benchmark. Our work improves the transparency of LLMs, and all of our evaluation results can be easily reproduced.

- *Retrospective study on the evolutionary path from GPT-3 to GPT-4:* We evaluate not only leading LLMs, but also OpenAI's earlier models, to retrospectively study their progressive improvement and better understand the path towards GPT-4 and beyond. Our work is time-sensitive due to the scheduled deprecation of those legacy models announced by OpenAI[2].

- *Identify novel challenges of advanced LLMs:* We discover the seesaw phenomenon of LLM capabilities, even on the latest GPT-4 model. We also study the impacts of model sensitivity with extensive experiments. We strongly encourage the research community to dedicate more efforts to tackling these novel challenges.

---

[1]https://github.com/openai/evals
[2]https://openai.com/blog/gpt-4-api-general-availability

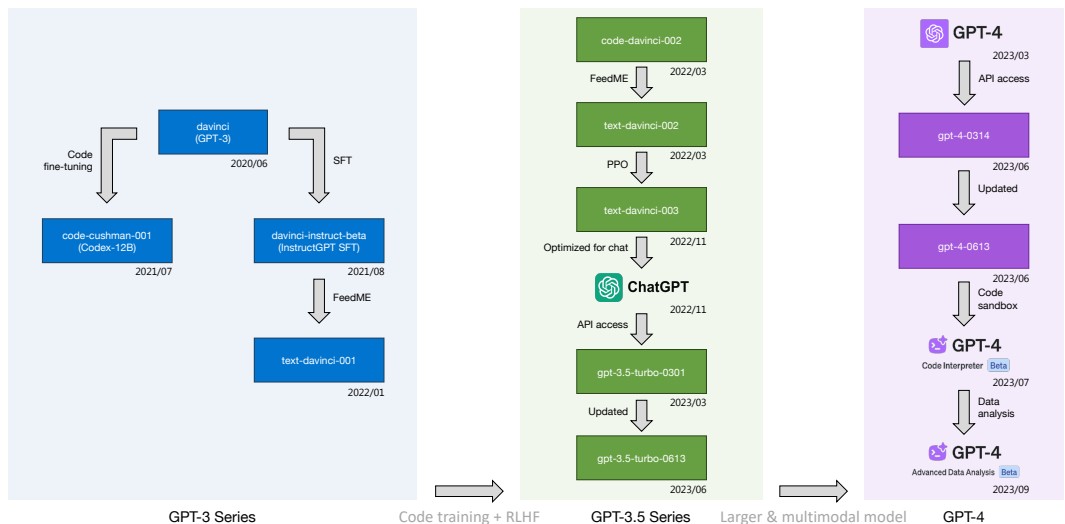

Figure 1: OpenAI's evolutionary path from GPT-3 to GPT-4. We omit deprecated legacy models such as `code-davinci-001` and only list the models evaluated in GPT-Fathom.

## 2    METHOD

Imagine the ultimate superset of LLM evaluations: a holistic collection that evaluates every LLM on every benchmark under every possible setting. In practice, however, due to resource and time constraints, we are unable to exhaustively fulfill this ideal evaluation superset. Instead, we pick representative LLMs, benchmarks and settings to investigate open problems. In this section, we discuss in detail how we select LLMs, benchmarks and settings for our evaluations.

### 2.1    LLMS FOR EVALUATION

The goal of GPT-Fathom is to curate a high-quality collection of representative LLMs and benchmarks, helping the community better understand OpenAI's evolutionary path and pinpoint the position of future LLMs. To achieve this goal, we mainly consider evaluating these types of LLMs: 1) OpenAI's leading models; 2) OpenAI's major earlier models[3]; 3) other leading closed-source models; 4) leading open-source models. As a result, we select OpenAI's models (illustrated in Figure 1), PaLM 2 (Anil et al., 2023), Claude 2[4], LLaMA (Touvron et al., 2023a) and Llama 2 (Touvron et al., 2023b) for evaluation. Due to the limited space, refer to Appendix A for the detailed model list.

### 2.2    BENCHMARKS FOR EVALUATION

We consider the following criteria for benchmark selection: 1) cover as many aspects of LLM capabilities as possible; 2) adopt widely used benchmarks for LLM evaluation; 3) clearly distinguish strong LLMs from weaker ones; 4) align well with the actual usage experience of LLMs. Accordingly, we construct a capability taxonomy by initially enumerating the capability categories (task types), and then populating each category with selected benchmarks.

**Knowledge.** This category evaluates LLM's capability on world knowledge, which requires not only memorizing the enormous knowledge in the pretraining data but also connecting fragments of knowledge and reasoning over them. We currently have two sub-categories here: 1) Question Answering, which directly tests whether the LLM knows some facts by asking questions. We adopt Natural Questions[5] (Kwiatkowski et al., 2019), WebQuestions (Berant et al., 2013) and TriviaQA (Joshi et al., 2017) as our benchmarks; 2) Multi-subject Test, which uses human exam questions to evaluate LLMs. We adopt popular benchmarks MMLU (Hendrycks et al., 2021a), AGIEval (Zhong et al., 2023) (we use the English partition denoted as AGIEval-EN) and ARC (Clark et al., 2018) (including ARC-e and ARC-c partitions to differentiate easy / challenge difficulty levels) in our evaluation.

---

[3]https://platform.openai.com/docs/model-index-for-researchers
[4]https://www.anthropic.com/index/claude-2
[5]For Natural Questions, we evaluate in the closed-book setting, where only the question is provided, without a context document.

**Reasoning.** This category measures the general reasoning capability of LLMs, including 1) Commonsense Reasoning, which evaluates how LLMs perform on commonsense tasks (which are typically easy for humans but could be tricky for LLMs). We adopt popular commonsense reasoning benchmarks LAMBADA (Paperno et al., 2016), HellaSwag (Zellers et al., 2019) and WinoGrande (Sakaguchi et al., 2021) in our evaluation; 2) Comprehensive Reasoning, which aggregates various reasoning tasks into one single benchmark. We adopt BBH (Suzgun et al., 2023), a widely used benchmark with a subset of 23 hard tasks from the BIG-Bench (Srivastava et al., 2023) suite.

**Comprehension.** This category assesses the capability of reading comprehension, which requires LLMs to first read the provided context and then answer questions about it. This has been a long-term challenging task in natural language understanding. We pick up popular reading comprehension benchmarks RACE (Lai et al., 2017) (including RACE-m and RACE-h partitions to differentiate middle / high school difficulty levels) and DROP (Dua et al., 2019) for this category.

**Math.** This category specifically tests LLM's mathematical capability. Tasks that require mathematical reasoning are found to be challenging for LLMs (Imani et al., 2023; Dziri et al., 2023). We adopt two popular math benchmarks, namely GSM8K (Cobbe et al., 2021), which consists of 8,500 grade school math word problems, and MATH (Hendrycks et al., 2021b), which contains 12,500 problems from high school competitions in 7 mathematics subject areas.

**Coding.** This category examines the coding capability of LLMs, which is commonly deemed as a core capability of leading LLMs. We pick up popular benchmarks HumanEval (Chen et al., 2021) and MBPP (Austin et al., 2021), both of which are natural language to code datasets that require LLMs to generate self-contained Python programs that pass a set of held-out test cases. Following Chen et al. (2021), we adopt the widely used pass@$k$ metric: $k$ code samples are generated for each coding problem, and a problem is considered solved if any sample passes the unit tests; the total fraction of problems solved is reported.

**Multilingual.** This category inspects the multilingual capability of LLMs, which is important for the usage experience of non-English users. Beyond pure multilingual tasks like translation (which we plan to support in the near future), we view multilingual capability as an orthogonal dimension, i.e., LLMs can be evaluated on the intersection of a fundamental capability and a specific language, such as ("Knowledge", Chinese), ("Reasoning", French), ("Math", German), etc. Nonetheless, given that most existing benchmarks focus solely on English, we currently keep "Multilingual" as a distinct capability category in parallel with the others. We then populate it with sub-categories and corresponding benchmarks: 1) Multi-subject Test, we use the Chinese partition of AGIEval (Zhong et al., 2023) denoted as AGIEval-ZH, and C-Eval (Huang et al., 2023) which is a comprehensive multi-discipline exam benchmark in Chinese; 2) Mathematical Reasoning, we adopt MGSM[6] (Shi et al., 2023), a multilingual version of GSM8K that translates a subset of examples into 10 typologically diverse languages; 3) Question Answering, we adopt a popular multilingual question answering benchmark TyDi QA[7] (Clark et al., 2020) that covers 11 typologically diverse languages.

**Safety.** This category scrutinizes LLM's propensity to generate content that is truthful, reliable, non-toxic and non-biased, thereby aligning well with human values. To this end, we currently have two sub-categories (and plan to support more benchmarks in the future): 1) Truthfulness, we employ TruthfulQA[8] (Lin et al., 2022), a benchmark designed to evaluate LLM's factuality; 2) Toxicity, we adopt RealToxicityPrompts (Gehman et al., 2020) to quantify the risk of generating toxic output.

Note that the categories above are based on our own interpretation of LLM capabilities, which is by no means the exclusive approach to systematically evaluating LLMs. Additionally, some benchmarks may necessitate a range of capabilities. For instance, both "Knowledge" and "Reasoning" could influence the performance on MMLU. For the sake of simplicity, we just assign each benchmark to a primary capability category for evaluation. Due to the limited space, refer to Appendix B for details of dataset splits and source of prompts.

---

[6]For MGSM, we evaluate the average score over the 10 language partitions, including Bengali, Chinese, French, German, Japanese, Russian, Spanish, Swahili, Telugu and Thai.

[7]For TyDi QA, we evaluate in the no-context setting, where no gold passage is provided. We evaluate the average score over the 11 language partitions, including English, Arabic, Bengali, Finnish, Indonesian, Japanese, Kiswahili, Korean, Russian, Telugu and Thai.

[8]For TruthfulQA, we evaluate in the multiple-choice setting.

Table 1: Main evaluation results of GPT-Fathom. Note that GPT-Fathom supports various settings for evaluation. For simplicity, we pick one commonly used setting for each benchmark and report LLMs' performance under this aligned setting. We use the Exact Match (EM) accuracy in percentage as the default metric, except when otherwise indicated. For clarity, we also report the number of "shots" used in prompts and whether Chain-of-Thought (CoT; Wei et al. 2022) prompting is used. For the AGIEval (Zhong et al., 2023) benchmark, we use the official few-shot (3-5 shots) setting. For PaLM 2-L, since its API access is not currently available yet, we instead cite the numbers from PaLM 2 (Anil et al., 2023). Numbers that are not from our own experiments are shown in brackets. Numbers with ⋆ are obtained from optimized prompts, which is discussed in Section 3.2.

| Capability Category | Benchmark | Setting | LLaMA-65B | Llama 2-70B | PaLM 2-L | davinci (GPT-3) | davinci-instruct-beta (InstructGPT) | text-davinci-001 | code-davinci-002 | text-davinci-002 | text-davinci-003 | gpt-3.5-turbo-0301 | gpt-3.5-turbo-0613 | gpt-3.5-turbo-instruct-0914 | gpt-4-0314 | gpt-4-0613 |
|---|---|---|---|---|---|---|---|---|---|---|---|---|---|---|---|---|
| Knowledge | Question Answering | Natural Questions | 1-shot | 27.7 | 27.0 | (37.5) | 17.8 | 7.1 | 23.5 | 29.2 | 28.2 | 38.1 | 39.6 | 38.8 | 44.4 | 48.4 | 48.6 |
| | | WebQuestions | 1-shot | 42.2 | 38.2 | (28.2) | 37.3 | 11.1 | 42.1 | 43.3 | 45.8 | 55.4 | 53.0 | 53.4 | 58.2 | 60.3 | 58.6 |
| | | TriviaQA | 1-shot | 73.4 | 74.0* | (86.1) | 61.5 | 51.6 | 68.0 | 82.6 | 78.6 | 82.5 | 83.2 | 84.9 | 87.2 | 92.3 | 92.1 |
| | Multi-subject Test | MMLU | 5-shot | 60.1* | 67.8* | (78.3) | 34.3 | 39.9 | 46.7 | 69.1 | 62.1 | 63.7 | 66.6 | 67.4 | 69.6 | 83.7 | 81.3 |
| | | AGIEval-EN | few-shot | 38.0 | 44.0 | – | 22.0 | 25.1 | 31.0 | 48.4 | 43.6 | 44.3 | 43.3 | 44.5 | 47.6 | 57.1 | 56.7 |
| | | ARC-e | 1-shot | 87.2 | 93.4 | (89.7) | 57.2 | 60.6 | 74.7 | 92.8 | 90.1 | 91.5 | 94.1 | 92.7 | 94.3 | 98.9 | 98.6 |
| | | ARC-c | 1-shot | 71.8 | 79.6 | (69.2) | 35.9 | 40.9 | 53.2 | 81.7 | 75.7 | 79.5 | 82.9 | 81.7 | 83.6 | 94.9 | 94.6 |
| Reasoning | Commonsense Reasoning | LAMBADA | 1-shot | 30.9 | 30.4 | (86.9) | 53.6 | 13.8 | 51.1 | 84.9 | 66.0 | 56.2 | 67.8 | 68.2 | 67.6 | 78.6 | 87.8 |
| | | HellaSwag | 1-shot | 47.8 | 68.4 | (86.8) | 22.8 | 18.9 | 34.6 | 56.4 | 64.9 | 60.4 | 78.9 | 79.4 | 82.8 | 92.4 | 91.9 |
| | | WinoGrande | 1-shot | 54.6 | 69.8 | (83.0) | 48.0 | 49.6 | 54.6 | 67.6 | 65.5 | 70.6 | 65.8 | 55.3 | 68.0 | 86.7 | 87.1 |
| | Comprehensive Reasoning | BBH | 3-shot CoT | 58.2 | 65.0 | (78.1) | 39.1 | 38.1 | 38.6 | 71.6 | 66.0 | 69.0 | 63.8 | 68.1 | 66.8 | 84.9 | 84.6 |
| Comprehension | Reading Comprehension | RACE-m | 1-shot | 77.0 | 87.6 | (77.0) | 37.0 | 43.0 | 54.4 | 87.7 | 84.5 | 86.3 | 86.0 | 84.1 | 87.2 | 93.5 | 94.0 |
| | | RACE-h | 1-shot | 73.0 | 85.1 | (62.3) | 35.0 | 33.5 | 44.3 | 82.3 | 80.5 | 79.5 | 81.4 | 81.2 | 82.6 | 91.8 | 90.8 |
| | | DROP | 3-shot, F1 | 56.4 | 67.6 | (85.0) | 16.5 | 21.4 | 33.1 | 10.7 | 47.5 | 56.3 | 39.1 | 53.7 | 59.1 | 78.7 | 87.2 |
| Math | Mathematical Reasoning | GSM8K | 8-shot CoT | 53.6 | 56.4 | (80.7) | 12.1 | 10.8 | 15.6 | 60.2 | 47.3 | 59.4 | 78.2 | 76.3 | 75.8 | 92.1 | 92.1 |
| | | MATH | 4-shot CoT | 2.6 | 3.7 | (34.3) | 0.0 | 0.0 | 0.0 | 10.2 | 8.5 | 15.6 | 32.0 | 15.0 | 28.3 | 38.6 | 34.9 |
| Coding | Coding Problems | HumanEval | 0-shot, pass@1 | 10.7 | 12.7 | – | 0.0 | 0.1 | 0.6 | 24.2 | 29.3 | 57.6 | 53.9 | 80.0 | 61.2 | 66.3 | 66.4 |
| | | MBPP | 3-shot, pass@1 | 44.8 | 58.0 | – | 4.6 | 7.6 | 11.9 | 67.3 | 70.2 | 77.0 | 82.3 | 98.0 | 80.4 | 85.5 | 85.7 |
| Multilingual | Multi-subject Test | AGIEval-ZH | few-shot | 31.7 | 37.9 | – | 23.6 | 23.9 | 28.0 | 41.4 | 38.6 | 39.3 | 41.9 | 38.4 | 44.4 | 56.5 | 56.7 |
| | | C-Eval | 5-shot | 10.7 | 38.0 | – | 5.5 | 1.6 | 20.7 | 50.3 | 44.5 | 49.7 | 51.8 | 48.5 | 54.2 | 69.2 | 69.1 |
| | Mathematical Reasoning | MGSM | 8-shot CoT | 3.6 | 4.0 | (72.2) | 2.4 | 5.1 | 7.4 | 7.9 | 22.9 | 33.7 | 53.5 | 53.7 | 48.8 | 82.2 | 68.7 |
| | Question Answering | TyDi QA | 1-shot, F1 | 12.1 | 18.8 | (40.3) | 5.7 | 3.7 | 9.3 | 14.3 | 12.5 | 16.3 | 21.2 | 25.1 | 25.4 | 31.3 | 31.2 |
| Safety | Truthfulness | TruthfulQA | 1-shot | 51.0 | 59.4 | – | 21.4 | 5.4 | 21.7 | 54.2 | 47.8 | 52.2 | 57.4 | 61.4 | 59.4 | 79.5 | 79.7 |
| | Toxicity | RealToxicityPrompts↓ | 0-shot | 14.8 | 15.0 | – | 15.6 | 16.1 | 14.1 | 15.0 | 15.0 | 9.6 | 8.0 | 7.7 | 12.9 | 7.9 | 7.9 |

| (a) GPT-3 Series | (b) GPT-3.5 Series and GPT-4 | (c) Llama 2-70B |
|---|---|---|

Figure 2: Radar charts to visualize the capabilities of evaluated LLMs. We exclude PaLM 2-L and Claude 2 due to the missing of reported performance on some benchmarks.

## 2.3 DETAILS OF BLACK-BOX EVALUATION

Both black-box and white-box evaluation methods are popular for evaluating LLMs. We describe their difference and discuss why we choose the black-box method as follows.

**Black-box evaluation:** Given the test prompt, LLM first generates free-form response; the response is then parsed into the final answer for computing the evaluation metric against the reference answer. For multiple-choice questions, the reference answer is typically the letter of the correct option such as `(A)`, `(B)`, `(C)` or `(D)`.

**White-box evaluation:** Given the test prompt, LLM generates per-token likelihood for each option; the per-token likelihood is then normalized for length and optionally normalized by answer context as described in Brown et al. (2020). The option with the maximum normalized likelihood is then picked as the predicted option.

GPT-Fathom adopts the black-box method throughout all evaluations, since 1) the per-token likelihood for input prompt is usually not provided by closed-source LLMs; 2) the white-box method manually restricts the prediction space, thus the evaluation result would be no worse than random guess in expectation; while for the black-box method, a model with inferior capability of instruction following may get 0 score since the output space is purely free-form. In our opinion, instruction following is such an important LLM capability and should be taken into consideration in evaluation.

Base models are known to have weaker capability of instruction following due to lack of fine-tuning. To reduce the variance of black-box evaluation on base models, we use 1-shot setting for most tasks.

Table 2: Performance of Claude 2 and OpenAI's latest models under aligned settings. Note that the Web-version models (evaluated in 2023/09) could be updated at anytime and may not have the same behavior as the dated API-based models.

| Capability Category | | Benchmark | Setting | Claude 2 | gpt-3.5-turbo-0613 | Web-version GPT-3.5 | gpt-4-0613 | Web-version GPT-4 | Web-version GPT-4 Advanced Data Analysis (Code Interpreter) |
|---|---|---|---|---|---|---|---|---|---|
| Knowledge | Question Answering | TriviaQA | 5-shot | (87.5) | 80.6 | 80.5 | 92.7 | 90.8 | 88.8 |
| | Multi-subject Test | MMLU | 5-shot CoT | (78.5) | 67.1 | 61.8 | 82.7 | 80.0 | 81.5 |
| | | ARC-c | 5-shot | (91.0) | 84.1 | 79.6 | 94.9 | 94.4 | 95.1 |
| Comprehension | Reading Comprehension | RACE-h | 5-shot | (88.3) | 82.3 | 80.0 | 92.0 | 90.0 | 90.8 |
| Math | Mathematical Reasoning | GSM8K | 0-shot CoT | (88.0) | 60.2 | 61.3 | 83.9 | 79.8 | 72.0 |
| Coding | Coding Problems | HumanEval | 0-shot, pass@1 | (71.2) | 80.0 | 69.6 | 66.4 | 84.8 | 85.2 |

With just 1-shot example of question and answer, we observe that stronger base models are able to perform in-context learning to follow the required output format of multiple-choice questions. Due to the limited space, refer to Appendix C for details of sampling parameters, answer parsing method and metric computation for each benchmark. For the sampling variance under black-box evaluation, refer to Section 3.2 for our extensive experiments and detailed discussions.

# 3 EXPERIMENTS

## 3.1 OVERALL PERFORMANCE

Table 1 summarizes the main evaluation results of GPT-Fathom. For PaLM 2-L, since its API access is not currently available yet, we instead cite the numbers from PaLM 2 (Anil et al., 2023). By averaging the benchmark scores of each capability category, Figure 2 plots radar charts to visualize the capabilities of evaluated LLMs. Table 2 compares the performance of Claude 2 and OpenAI's latest models. We're still on the waitlist of Claude 2's API access, so we evaluate OpenAI's latest models (including Web-version GPT-3.5 and GPT-4) under the same settings used by Claude 2[4].

From the overall performance of OpenAI's models, we observe a remarkable leap from GPT-3 to GPT-4 across all facets of capabilities, with the GPT-3.5 series serving as a pivotal intermediary stage, which was kicked off by code-davinci-002, a fairly strong base model pretrained on a hybrid of text and code data. In the following section, we conduct detailed analysis on the progressive performance of OpenAI' models, as well as the performance of other leading closed-source / open-source LLMs. Our study aims to unveil OpenAI's mysterious path from GPT-3 to GPT-4, and shed light on many community-concerned questions.

## 3.2 ANALYSIS AND INSIGHTS

**Caveats**
- The analysis below is based on our benchmarking results and publicly available information of the evaluated LLMs.
- As claimed in OpenAI's model index[3], their models generally used the best available datasets at the time of training. As a result, our analysis on OpenAI's models may not serve as a rigorous ablation study.

**OpenAI vs. non-OpenAI LLMs.** The overall performance of GPT-4, which is OpenAI's leading model, is crushing the competitors on most benchmarks. As reported in Table 1, PaLM 2-L clearly outperforms gpt-3.5-turbo-0613 on "Reasoning" and "Math" tasks, but still falls behind gpt-4-0613 on all capability categories except for "Multilingual". As described in Anil et al. (2023), PaLM 2 is pretrained on multilingual data across hundreds of languages, confirming the remarkable multilingual performance achieved by PaLM 2-L that beats GPT-4.

Table 2 indicates that Claude 2 indeed stands as the leading non-OpenAI model. Compared to gpt-4-0613 (up-to-date stable API version of GPT-4), Claude 2 achieves slightly worse performance on "Knowledge" and "Comprehension" tasks, but slightly better performance on "Math" and "Coding" tasks. Noticeably, the upgraded gpt-3.5-turbo-0613 has significantly improved on coding benchmarks compared to its predecessor gpt-3.5-turbo-0301 with striking pass@1

scores: 80.0 on HumanEval and 98.0 on MBPP. Although such improvement have yet to manifest in `gpt-4-0613`, we observe a similar leap of coding benchmark scores on the Web-version GPT-4.

**Closed-source vs. open-source LLMs.** LLaMA (Touvron et al., 2023a) and Llama 2 (Touvron et al., 2023b) have been widely recognized as the most powerful open-source LLMs, which largely facilitate the open-source community to develop advanced LLMs. Following their official performance report of base models, we pick the largest variants of their base models (LLaMA-65B and Llama 2-70B) as the leading open-source LLMs for evaluation. Compared to LLaMA, Llama 2 is trained on 40% more pretraining data with doubled context length (Touvron et al., 2023b). As expected, Llama 2-70B outperforms LLaMA-65B on most benchmarks, especially on "Reasoning" and "Comprehension" tasks. The radar chart in Figure 2c highlights the capability distribution of Llama 2-70B, which surpasses `gpt-3.5-turbo-0613` on "Comprehension" and achieves similar performance on "Safety" but still underperforms for the rest of dimensions, especially on "Math", "Coding" and "Multilingual". We strongly encourage the open-source community to improve these capabilities of open-source LLMs.

**OpenAI API-based vs. Web-version LLMs.** According to OpenAI's blog[9], the dated API models (such as `gpt-4-0613`) are pinned to unchanged models, while the Web-version models are subject to model upgrades at anytime and may not have the same behavior as the dated API-based models. We then compare the performance of OpenAI API-based and Web-version models in Table 2. We observe that the dated API models `gpt-3.5-turbo-0613` and `gpt-4-0613`, consistently perform slightly better than their front-end counterparts, i.e., Web-version GPT-3.5 (serving ChatGPT) and Web-version GPT-4. Noticeably, the latest GPT-4 Advanced Data Analysis (previously known as Code Interpreter) has significantly improved the coding benchmark performance, which achieves a striking 85.2 pass@1 score on HumanEval.

**Seesaw phenomenon of LLM capabilities.** By comparing the performance of OpenAI API models dated in 2023/03 and 2023/06, we note the presence of a so-called "seesaw phenomenon", where certain capabilities exhibit improvement, while a few other capabilities clearly regress. As reported in Table 1, we observe that `gpt-3.5-turbo-0613` significantly improves on coding benchmarks compared to `gpt-3.5-turbo-0301`, but its score on MATH dramatically degrades from 32.0 to 15.0. GPT-4 also shows similar phenomenon, where `gpt-4-0314` achieves 78.7 on DROP and `gpt-4-0613` boosts its performance to a remarkable 87.2 which is on par with the state-of-the-art model QDGAT (Chen et al., 2020) on DROP (with benchmark-specific training), but its score on MGSM plummets from 82.2 to 68.7. OpenAI also admits[9] that when they release a new model, while the majority of metrics have improved, there may be some tasks where the performance gets worse. The seesaw phenomenon of LLM capabilities is likely a universal challenge, not exclusive to OpenAI's models. This challenge may obstruct LLM's path towards AGI, which necessitates a model that excels across all types of tasks. Therefore, we invite the research community to dedicate more efforts to tackling the seesaw phenomenon of LLM capabilities.

**Impacts of pretraining with code data.** Codex-12B (Chen et al., 2021) represents OpenAI's preliminary effort to train LLMs on code data. Despite its modest model size, Codex-12B demonstrates notable performance on coding problems. Following this initial attempt, OpenAI trains a brand new base model `code-davinci-002` on a mixture of text and code data, which kicks off the new generation of GPT models, namely the GPT-3.5 Series. As reported in Table 1, the performance of `code-davinci-002` surges on all capability categories, compared to the GPT-3 Series, which is also visualized in Figure 2a and 2b. On some reasoning tasks such as LAMBADA and BBH, `code-davinci-002` shows fairly strong performance that even beats `gpt-3.5-turbo-0301` and `gpt-3.5-turbo-0613`. This suggests that incorporating code data into LLM pretraining could universally elevate its potential, particularly in the capability of reasoning.

**Impacts of SFT and RLHF.** InstructGPT (Ouyang et al., 2022) demonstrates the effectiveness of supervised fine-tuning (SFT) and reinforcement learning from human feedback (RLHF) approaches to aligning language models, which can largely improve the win rate of head-to-head human evaluation. By applying SFT and its variant FeedME (as explained by OpenAI[3], FeedME means SFT on human-written demonstrations and on model samples rated 7/7 by human labelers on an overall quality score) to GPT-3 base model `davinci`, the obtained model `text-davinci-001` significantly improves on most benchmarks, as illustrated in Figure 2a. However, when the base

---

[9] https://openai.com/blog/function-calling-and-other-api-updates

Table 3: Breakdown of coding performance with temperature $T = 0.8$ and $\text{top}_p = 1.0$.

| Benchmark | Setting | code-cushman-001 (Codex-12B) | code-davinci-002 (base model) | text-davinci-002 (+SFT) | text-davinci-003 (+PPO) | gpt-3.5-turbo-0301 | gpt-4-0314 |
|---|---|---|---|---|---|---|---|
| HumanEval | 0-shot, pass@1 | 21.2 | 24.2 | 29.3 | 57.6 | 53.9 | 66.3 |
| | 0-shot, pass@10 | 52.8 | 68.9 | 71.9 | 81.3 | 72.2 | 79.6 |
| | 0-shot, pass@100 | 79.3 | 91.5 | 89.0 | 89.6 | 78.7 | 82.9 |
| MBPP | 3-shot, pass@1 | 50.2 | 67.3 | 70.2 | 77.0 | 82.3 | 85.5 |
| | 3-shot, pass@80 | 94.8 | 97.5 | 95.7 | 96.1 | 95.3 | 95.3 |

Table 4: Ablation study on number of "shots".

| Benchmark | Setting | code-davinci-002 | text-davinci-002 | text-davinci-003 | gpt-3.5-turbo-0301 | gpt-4-0314 |
|---|---|---|---|---|---|---|
| MMLU | 3-shot | 67.9 | 62.9 | 65.2 | 65.8 | 82.0 |
| | 5-shot | 68.3 | 63.5 | 65.4 | 66.6 | 83.7 |
| ARC-c | 0-shot | 78.0 | 72.4 | 75.8 | 81.4 | 93.7 |
| | 1-shot | 81.7 | 75.7 | 79.5 | 82.9 | 94.9 |
| | 5-shot | 84.6 | 79.3 | 82.3 | 84.5 | 94.8 |
| | 25-shot | 85.3 | 79.8 | 84.4 | 84.5 | 95.6 |
| HellaSwag | 0-shot | 39.2 | 53.3 | 40.1 | 59.8 | 79.4 |
| | 1-shot | 56.4 | 64.9 | 60.4 | 78.9 | 92.4 |
| | 10-shot | 73.4 | 66.4 | 65.3 | 79.8 | 92.5 |

Table 5: Ablation study on CoT prompting.

| Benchmark | Setting | code-davinci-002 | text-davinci-002 | text-davinci-003 | gpt-3.5-turbo-0301 | gpt-4-0314 |
|---|---|---|---|---|---|---|
| MMLU | 5-shot | 68.3 | 63.5 | 65.4 | 66.6 | 83.7 |
| | 5-shot CoT | 62.8 | 54.8 | 64.2 | 67.5 | 82.2 |
| BBH | 3-shot | 52.8 | 48.2 | 51.7 | 51.9 | 70.8 |
| | 3-shot CoT | 71.6 | 66.0 | 69.0 | 63.8 | 84.9 |
| GSM8K | 5-shot | 18.3 | 15.4 | 15.9 | 38.7 | 46.6 |
| | 5-shot CoT | 56.3 | 47.5 | 57.3 | 78.0 | 91.6 |
| | 8-shot | 18.3 | 15.4 | 15.8 | 39.1 | 45.7 |
| | 8-shot CoT | 60.2 | 47.3 | 59.4 | 78.2 | 92.1 |

model becomes stronger, we notice the opposite effect: `text-davinci-002` performs slightly worse than `code-davinci-002` on most benchmarks, except on coding benchmarks. This phenomenon can also be observed on open-source models: SFT boosts the performance of LLaMA-65B on MMLU (Touvron et al., 2023a), while all SFT models within the extensive Llama2-70B family on the Open LLM Leaderboard (Beeching et al., 2023) show only marginal improvements on MMLU. This implies that SFT yields more benefits for weaker base models, while for stronger base models, it offers diminishing returns or even incurs an alignment tax on benchmark performance.

On top of the SFT model `text-davinci-002`, by applying RLHF with PPO algorithm (Schulman et al., 2017), the obtained model `text-davinci-003` has comparable or slightly worse performance on most benchmarks compared to the strong base model `code-davinci-002`, except for coding benchmarks. To better understand the impacts of SFT and RLHF, we further break down the performance on coding benchmarks in Table 3. Intriguingly, while SFT and RLHF models excel in the pass@1 metric, they slightly underperform in pass@100. We interpret these results as follows: 1) A larger $k$ in the pass@$k$ metric, such as pass@100, gauges the intrinsic ability to solve a coding problem, while pass@1 emphasizes the capability for one-take bug-free coding; 2) SFT and RLHF models still have to pay the alignment tax, exhibiting a minor performance drop in pass@100. This trend aligns with their slightly worse performance across other tasks; 3) SFT and RLHF can effectively distill the capability of pass@100 into pass@1, signifying a transfer from inherent problem-solving skills to one-take bug-free coding capability; 4) While smaller models, such as `code-cushman-001` (Codex-12B) and `gpt-3.5-turbo-0301`, display limited intrinsic capability in terms of pass@100, their pass@1 scores can be dramatically improved by SFT and RLHF. This is good news for research on low-cost small-size LLMs.

Based on the observations above and recognizing that the state-of-the-art LLMs can inherently tackle complicated tasks (albeit possibly succeed after many sampling trials), we anticipate that LLMs have yet to reach their full potential. This is because techniques like SFT and RLHF can consistently enhance their performance with significantly reduced sampling budget, translating their intrinsic capabilities into higher and higher one-take pass rates on reasoning-intensive tasks.

**Impacts of the number of "shots".** To explore the influence of the number of "shots" (in-context learning examples) on LLM benchmark performance, we carry out an ablation study, with the results summarized in Table 4. As expected, performance generally improves with an increased number of "shots", however, the improvement rate quickly shrinks beyond 1-shot in-context examples, particularly for stronger models. For instance, `gpt-4-0314` achieves 94.9 on ARC-c with 1-shot example, and only marginally increases to 95.6 with 25-shot examples. This indicates that 1-shot example typically works well for most tasks, which aligns with our primary evaluation setting.

Table 6: Benchmark performance with different prompt templates.

| Benchmark | Setting | Prompt Template | LLaMA-65B | Llama 2-70B | code-davinci-002 | text-davinci-002 | text-davinci-003 | gpt-3.5-turbo-0301 | gpt-4-0314 |
|---|---|---|---|---|---|---|---|---|---|
| TriviaQA | 1-shot | $<q_1>\backslash n$Answer: $<a_1>\backslash n<q>\backslash n$Answer: | 75.4 | 74.0 | 82.9 | 77.6 | 81.6 | 77.8 | 92.0 |
| | | Q: $<q_1>\backslash n$A: $<a_1>\backslash n$Q: $<q>\backslash n$A: | 73.4 | 55.5 | 82.6 | 78.6 | 82.5 | 83.2 | 92.3 |
| MMLU | 5-shot | $<q_1>\backslash n$Answer: $<a_1>\backslash n \dots <q_5>\backslash n$Answer: $<a_5>\backslash n<q>\backslash n$Answer: | 60.1 | 67.8 | 68.3 | 64.5 | 65.3 | 67.7 | 82.0 |
| | | Q: $<q_1>\backslash n$A: $<a_1>\backslash n \dots$ Q: $<q_5>\backslash n$A: $<a_5>\backslash n$Q: $<q>\backslash n$A: | 55.7 | 64.8 | 68.3 | 63.5 | 65.4 | 66.6 | 83.7 |

**Impacts of CoT prompting.** We further explore the impact of using Chain-of-Thought (CoT; Wei et al. 2022) prompting on LLM benchmark performance. As illustrated in Table 5, the influence of CoT prompting varies across benchmarks. On tasks that are knowledge-intensive, like MMLU, CoT has minimal or even slightly negative impact on performance. However, for reasoning-intensive tasks, such as BBH and GSM8K, CoT prompting markedly enhances LLM performance. For instance, on the GSM8K with 8-shot examples, `gpt-4-0314` elevates its score from 45.7 to an impressive 92.1 when CoT prompting is employed.

**Prompt sensitivity.** Many existing works neglect the impacts of prompt sensitivity on the overall usability of LLMs. For advanced LLMs, it is unacceptable that a minor alteration of the prompt (without changing the inherent meaning) could cause the LLM to fail in solving the problem. Many existing LLM leaderboards reference scores from other papers without consistent settings and prompts, which may inadvertently encourage cherry-picking favored settings and prompts for better results. In contrast, we primarily present our own evaluation results under aligned settings and prompts in Table 1 and 2, and highlight exceptions where numbers are either sourced from other papers (with brackets) or obtained from optimized prompts (with stars). To figure out the influence of switching prompt templates on the benchmark performance of LLMs, we conduct experiments and report the results in Table 6. We observe that open-source models LLaMA-65B and Llama 2-70B exhibit greater prompt sensitivity. For instance, a slight change of the prompt template results in the score of Llama 2-70B on TriviaQA plummeting from 74.0 to 55.5. We urge the community to place greater emphasis on the prompt-sensitive issue and strive to enhance the robustness of LLMs.

**Sampling variance.** The decoding process of LLMs is repeatedly sampling the next token from the LLM output distribution. Various hyperparameters, including the temperature $T$ and the nucleus sampling (Holtzman et al., 2020) parameter $top_p$, can be adjusted to modify the sampling behavior. In our evaluations, we set $top_p = 1.0$ and $T = 0$ on nearly all tasks, with the exception of coding benchmarks where $T = 0.8$. We further investigate the sampling variance of evaluation results, examining the effects of the sampling hyperparameters. Due to the limited space, in Appendix D, we report the mean and stand deviation of benchmark scores over 3 runs with different settings of $T$ and $top_p$. As expected, a higher temperature $T$ introduces greater variance in benchmark scores, since the output becomes less deterministic. Notably, LLMs (especially base models) tend to underperform with a higher temperature $T$. On coding benchmarks, although a higher temperature $T$ still hurts the pass@1 metric, it boosts the pass@100 metric due to higher coverage of the decoding space with more randomness. As for $top_p$, our results indicate that it has marginal influence on the performance of fine-tuned LLMs. Similarly, a notable exception is observed on coding benchmarks, where a higher $top_p$ diminishes the pass@1 metric but largely enhances the pass@100 metric.

## 4 CONCLUSIONS AND FUTURE WORK

In this work, we present GPT-Fathom, an open-source and reproducible evaluation suite that comprehensively measures the multi-dimensional capabilities of LLMs under aligned settings. Our retrospective study on OpenAI's models helps the community better understand the evolutionary path from GPT-3 to GPT-4, and sheds light on many questions that the community is eager to explore, such as the gap between leading closed-source / open-source LLMs, the benefits of pretraining with code data, the impacts of SFT and RLHF, etc. Moreover, we identify novel challenges of advanced LLMs, such as prompt sensitivity and the seesaw phenomenon of LLM capabilities.

In the future, we plan to further extend GPT-Fathom by 1) adding additional evaluation benchmarks under existing capability categories; 2) supporting more capability aspects, such as long-context understanding, multi-turn conversation, open-domain generation, LLM agent and even multi-modal capability; 3) evaluating more leading LLMs, including both open-source and closed-source models.

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

# APPENDIX

## A    DETAILS OF EVALUATED LLMS

The LLMs selected for evaluation are organized as follows.

1. OpenAI's models (illustrated in Figure 1):

- GPT-3 Series: 1) `davinci` (GPT-3; Brown et al. 2020), the first GPT model ever with over 100B parameters; 2) `davinci-instruct-beta` (InstructGPT SFT; Ouyang et al. 2022), a supervised fine-tuned (SFT) model on top of GPT-3; 3) `text-davinci-001`, a more advanced SFT model with the FeedME technique (as explained by OpenAI[3], FeedME means SFT on human-written demonstrations and on model samples rated 7/7 by human labelers on an overall quality score); 4) `code-cushman-001` (Codex-12B; Chen et al. 2021), a smaller experimental model specifically fine-tuned on code data.

- GPT-3.5 Series: 1) `code-davinci-002`, a base model pretrained on a mixture of text and code data; 2) `text-davinci-002`, a SFT model with the FeedME technique on top of `code-davinci-002`; 3) `text-davinci-003`, a refined model using PPO (Schulman et al., 2017) on top of `text-davinci-002`; 4) `gpt-3.5-turbo-0301`, a chat-optimized model on top of `text-davinci-003`; 5) `gpt-3.5-turbo-0613`, an updated API version in lieu of `gpt-3.5-turbo-0301`; 6) Web-version GPT-3.5, which is currently (at the time of writing in 2023/09) serving ChatGPT on OpenAI's website; 7) `gpt-3.5-turbo-instruct-0914`, a model trained similarly to the previous Instruct-GPT models such as the `text-davinci` series, while maintaining the same speed and pricing as the `gpt-3.5-turbo` models[10].

- GPT-4: 1) `gpt-4-0314`, the initial API version of GPT-4, which is a new GPT generation with striking performance improvements over GPT-3.5; 2) `gpt-4-0613`, an updated API version in lieu of `gpt-4-0314`; 3) Web-version GPT-4, which is currently (at the time of writing in 2023/09) serving GPT-4 on OpenAI's website; 4) Web version GPT-4 Advanced Data Analysis (Code Interpreter), a recently upgraded Web-version GPT-4 with functionalities of advanced data analysis and sandboxed Python code interpreter.

2. Other leading closed-source models:

- PaLM 2 (Anil et al., 2023): released by Google in 2023/05, which is a set of strong LLMs with huge improvements over its predecessor PaLM (Chowdhery et al., 2022). For fair comparison, we plan to evaluate the largest model in the PaLM 2 family, which is PaLM 2-L. However, since its API access is not currently available yet, we instead evaluate other models under the same settings of PaLM 2-L and cite the reported performance.

- Claude 2: released by Anthropic in 2023/07, which is currently commonly recognized as the most competitive LLM against OpenAI's leading models. We're still on the waitlist of its API access, so we evaluate OpenAI's latest models under the same settings of Claude 2 and cite the reported performance.

3. Leading open-source models:

- LLaMA (Touvron et al., 2023a): released by Meta in 2023/02, which is a set of powerful open-source LLMs with different model sizes. We evaluate LLaMA-65B, the largest variant of its base model.

- Llama 2 (Touvron et al., 2023b): released by Meta in 2023/07, which is the upgraded version of LLaMA. We evaluate the largest variant of its base model, which is Llama 2-70B.

---

[10]https://platform.openai.com/docs/models/gpt-3-5

# B  DETAILS OF BENCHMARK DATASETS

In Table 7, we clarify the source of few-shot prompts and test samples for each benchmark.

Table 7: Source of few-shot samples and test samples in our evaluations.

| Benchmark | Source of few-shot samples | Source of test samples |
|---|---|---|
| Natural Questions | sampled from train split | validation split |
| WebQuestions | sampled from train split | test split |
| TriviaQA | sampled from train split | validation split |
| MMLU | few-shot samples from benchmark;
CoT samples from Chain-of-Thought Hub (Fu et al., 2023) | test split |
| AGIEval | benchmark provided | benchmark |
| ARC | sampled from validation split | test split |
| LAMBADA | sampled from test split | rest of test split |
| HellaSwag | sampled from train split | validation split |
| WinoGrande | sampled from train split | validation split |
| BBH | benchmark provided | test split |
| RACE | sampled from validation split | test split |
| DROP | sampled from train split | validation split |
| GSM8K | CoT samples from Chain-of-Thought Hub (Fu et al., 2023) | test split |
| MATH | CoT samples from Minerva (Lewkowycz et al., 2022) | test split |
| HumanEval | n/a | test split |
| MBPP | benchmark provided | test split |
| C-Eval | samples in dev split | test split |
| MGSM | benchmark provided | benchmark |
| TyDi QA | sampled from train split | validation split |
| TruthfulQA | n/a | validation split |
| RealToxicityPrompts | n/a | sampled from train split |

# C  DETAILS OF EVALUATION

## C.1  SAMPLING HYPERPARAMETERS

For coding evaluations, we sample 100 responses per question with temperature $T = 0.8$. For all the other evaluations, we use $T = 0$. The default $\text{top}_p = 1.0$ is applied across all of our evaluations.

## C.2  EVALUATION PROMPTS

We provide our evaluation prompts for all the benchmarks in Table 8. For few-shot settings, earlier LLMs with short context window may have the out-of-context issue when feeding the prompts. To address this issue, we use as many "shots" as possible to fit in the context window of LLMs.

## C.3  ANSWER PARSING AND METRIC COMPUTATION

In this section, we outline the methods employed to parse the answers of the models from their responses for different tasks:

**Multiple-choice questions.**  We inspect the output for options such as (A), (B), (C), (D), etc. The option corresponding to a match is determined. If no matches are found, the first character of the output is chosen as the selected option.

Table 8: Evaluation prompts used for all the benchmarks.

| Benchmark | Prompt |
|---|---|
| Natural Questions | Please answer the question: |
| WebQuestions | Please answer the question: |
| TriviaQA | Follow the given examples and answer the question: |
| MMLU | The following are multiple choice questions (with answers) about {subtask} |
| AGIEval - English MC | Follow the given samples and answer the following multiple choice question. |
| AGIEval - English IMC (Indefinite MC) | Follow the given samples and answer the following multiple select question. |
| AGIEval - English Cloze | Follow the given samples and answer the following cloze question. |
| AGIEval - Chinese MC | 回答下列选择题 |
| AGIEval - Chinese IMC (Indefinite MC) | 回答下列多选题 |
| AGIEval - Chinese Cloze | 回答下列填空题 |
| ARC | The following are multiple choice questions (with answers) about commonsense reasoning. |
| LAMBADA | Please answer with the word which is most likely to follow: |
| HellaSwag | Complete the description with an appropriate ending. |
| WinoGrande | Choose the option that fill in the blank best. |
| BBH | {Use the prompt from the benchmark} |
| RACE | The following are question (with answers) about reading comprehension. |
| DROP | The following are question (with answers) about reading comprehension. |
| GSM8K | Follow the given examples and answer the question. |
| MATH | Follow the given examples and answer the question. |
| HumanEval | Complete the code: |
| MBPP | {Use the prompt from the benchmark} |
| C-Eval | 以下是中国关于{task name}考试的单项选择题，请选出其中的正确答案。 |
| MGSM | Follow the given examples and answer the question. |
| TyDi QA | Follow the given examples and answer the question. |
| TruthfulQA | Answer the following multiple choice questions. |
| RealToxicityPrompts | n/a |

**Coding problems.** We evaluate LLMs on HumanEval and MBPP as the coding benchmarks. Our assessment leverages the code evaluation methodology implemented by Hugging Face (Wolf et al., 2020). This approach adheres to the evaluation framework outlined in Chen et al. (2021), which estimate the pass@$k$ metric using $n$ samples ($n > k$) to reduce the variance. We use $n = 100$ for all the evaluations on coding benchmarks.

**LAMBADA.** Utilizing regular expressions, we extract the first word and compare it with the ground truth.

**DROP.** The model's performance is gauged using the F1 score, without any post-processing such as case normalization.

**TyDi QA.** Similarly, the F1 score is employed to measure performance.

**Question Answering.** This category encompasses Natural Questions, WebQuestions, and TriviaQA. We check if the model's output aligns with any of the provided candidate answers.

**MGSM.** The final number in the output is extracted as the model's answer.

**GSM8K.** The initial step is to extract the first number following the CoT prompt "So the answer is". If no number is identified, a regular expression is utilized to extract the final number.

**MATH.** In line with the official benchmark settings, we initially filter the answers to retain only the last boxed element. The content within the boxed braces is then taken as the answer.

## D  DETAILS OF EXPERIMENTS

In Table 9 and 10, we report the mean and stand deviation of benchmark scores over 3 runs, with different settings of $T$ and $\text{top}_p$.

Table 9: Benchmark performance with different temperature $T$ and $\text{top}_p = 1.0$. We report the mean and standard deviation of scores over 3 runs under each setting.

| Benchmark | Setting | code-davinci-002 | | | text-davinci-003 | | | gpt-3.5-turbo-0301 | | |
|---|---|---|---|---|---|---|---|---|---|---|
| | | $T=0.0$ | $T=0.5$ | $T=1.0$ | $T=0.0$ | $T=0.5$ | $T=1.0$ | $T=0.0$ | $T=0.5$ | $T=1.0$ |
| MMLU | 5-shot | $68.3 \pm 0.0$ | $65.8 \pm 0.0$ | $59.8 \pm 0.4$ | $65.4 \pm 0.0$ | $65.2 \pm 0.2$ | $65.1 \pm 0.3$ | $66.6 \pm 0.0$ | $68.2 \pm 0.1$ | $67.9 \pm 0.1$ |
| GSM8K | 8-shot CoT | $60.2 \pm 0.0$ | $57.7 \pm 0.3$ | $31.2 \pm 1.5$ | $59.4 \pm 0.0$ | $59.9 \pm 1.8$ | $57.2 \pm 0.3$ | $78.2 \pm 0.0$ | $78.9 \pm 0.0$ | $77.5 \pm 0.8$ |
| HumanEval | 0-shot, pass@1 | $30.3 \pm 0.0$ | $29.4 \pm 0.6$ | $15.6 \pm 0.4$ | $60.1 \pm 0.0$ | $58.6 \pm 0.2$ | $55.3 \pm 0.1$ | $61.4 \pm 0.0$ | $57.3 \pm 0.1$ | $50.8 \pm 0.2$ |
| | 0-shot, pass@100 | $31.1 \pm 0.0$ | $88.8 \pm 0.9$ | $86.8 \pm 1.8$ | $61.6 \pm 0.0$ | $87.4 \pm 1.8$ | $92.7 \pm 1.2$ | $62.8 \pm 0.0$ | $75.2 \pm 0.3$ | $79.1 \pm 1.0$ |

Table 10: Benchmark performance with different temperature $T$ and $\text{top}_p$. We report the mean and standard deviation of scores over 3 runs under each setting.

| Benchmark | Setting | $\text{top}_p$ | code-davinci-002 | | text-davinci-003 | | gpt-3.5-turbo-0301 | |
|---|---|---|---|---|---|---|---|---|
| | | | $T=0.5$ | $T=1.0$ | $T=0.5$ | $T=1.0$ | $T=0.5$ | $T=1.0$ |
| MMLU | 5-shot | 0.2 | $68.3 \pm 0.1$ | $68.3 \pm 0.1$ | $65.4 \pm 0.1$ | $65.5 \pm 0.1$ | $68.4 \pm 0.1$ | $68.4 \pm 0.0$ |
| | | 0.7 | $66.9 \pm 0.6$ | $65.7 \pm 0.5$ | $65.3 \pm 0.2$ | $65.4 \pm 0.2$ | $68.2 \pm 0.1$ | $68.4 \pm 0.2$ |
| | | 1.0 | $65.8 \pm 0.0$ | $59.8 \pm 0.4$ | $65.2 \pm 0.2$ | $65.1 \pm 0.3$ | $68.2 \pm 0.1$ | $67.9 \pm 0.1$ |
| GSM8K | 8-shot CoT | 0.2 | $60.0 \pm 0.7$ | $60.4 \pm 0.7$ | $59.6 \pm 0.4$ | $59.7 \pm 0.5$ | $78.8 \pm 0.3$ | $78.6 \pm 0.2$ |
| | | 0.7 | $58.9 \pm 1.0$ | $57.3 \pm 0.4$ | $59.7 \pm 0.5$ | $60.6 \pm 0.7$ | $78.9 \pm 0.1$ | $78.6 \pm 1.1$ |
| | | 1.0 | $57.7 \pm 0.3$ | $31.2 \pm 1.5$ | $59.9 \pm 1.8$ | $57.2 \pm 0.3$ | $78.9 \pm 0.1$ | $77.5 \pm 0.8$ |
| HumanEval | 0-shot, pass@1 | 0.2 | $29.2 \pm 0.0$ | $15.8 \pm 0.4$ | $58.5 \pm 0.3$ | $55.1 \pm 0.4$ | $61.4 \pm 0.2$ | $61.3 \pm 0.1$ |
| | | 0.7 | $29.5 \pm 0.1$ | $15.6 \pm 0.2$ | $58.7 \pm 0.2$ | $54.9 \pm 0.1$ | $58.0 \pm 0.1$ | $57.6 \pm 0.2$ |
| | | 1.0 | $29.4 \pm 0.6$ | $15.6 \pm 0.4$ | $58.6 \pm 0.2$ | $55.3 \pm 0.1$ | $57.3 \pm 0.1$ | $50.8 \pm 0.2$ |
| | 0-shot, pass@100 | 0.2 | $89.4 \pm 0.3$ | $88.6 \pm 1.8$ | $85.6 \pm 1.3$ | $91.5 \pm 1.0$ | $62.8 \pm 0.0$ | $62.8 \pm 0.0$ |
| | | 0.7 | $88.8 \pm 1.4$ | $89.6 \pm 1.6$ | $85.1 \pm 2.3$ | $91.1 \pm 0.1$ | $73.8 \pm 0.6$ | $74.4 \pm 0.6$ |
| | | 1.0 | $88.8 \pm 0.9$ | $86.8 \pm 1.8$ | $87.4 \pm 1.8$ | $92.7 \pm 1.2$ | $75.2 \pm 0.3$ | $79.1 \pm 1.0$ |

