# OpenReview forum: "GPT-Fathom: Benchmarking Large Language Models to Decipher the Evolutionary Path towards GPT-4 and Beyond"
_ICLR.cc/2024/Conference — ICLR 2024 Conference Withdrawn Submission_

### Official Review · Reviewer_ThD7 · 2023-10-22

**Soundness:** 3 good
**Presentation:** 4 excellent
**Contribution:** 3 good
**Rating:** 6
**Confidence:** 4

**Summary:**

This work comprehensively evaluates 10+ leading LLMs, such as OpenAI's GPT series models, Claude 2, and Llama2, on 20+ curated benchmarks across 7 carefully chosen capability categories, including knowledge, reasoning, comprehension, math, code, multilingual, and safety. The comparison results offer valuable insights into the evolutionary path from GPT-3 series to GPT-3.5 series and GPT-4, and partially answer some important questions that are of curiosity to the community.

**Strengths:**

**Significance**: This work provides a high-quality and comprehensive benchmark for LLMs research, which may provide a good foundation for LLMs development and comparison.

**Quality**:  Each dimension of the evaluation benchmark (e.g., metric, model, used prompt, black-box evaluation vs white box evaluation, etc) is carefully chosen. The analysis about the evaluation results is well conducted and deliver some useful information.


**Clarity**: The paper is well-written, and the structure and figure is very clear.

**Weaknesses:**

**Originality**: l have seen that the authors clearly compare this work with previous LLMs benchmark work (in the penultimate paragraph in the introduction section), and it appears to be the first benchmark that consistently evaluates so many LLM models across multiple capability dimensions. I am curious to know if there are any novel evaluation dimensions proposed in this work.

**Questions:**

1. Have the authors open-sourced the benchmark work, and what costs are associated with evaluating a newly trained model?
2. Are the current capability dimensions sufficient to systematically evaluate current Low-Level Models, or is there any important metric that this work is missing?

---

> ### Author Response · Authors · 2023-11-19
> **Official Response to Reviewer ThD7**
>
> Thank you for your insightful assessment of our work and the constructive queries raised. We are delighted to hear that you found our paper to be of good quality, significance and clarity.
>
> **Response #1 - Originality**
>
> We appreciate your feedback on the originality of evaluation dimensions. The focus of our work is to provide insights for better understanding the key techniques of leading LLMs, so as to improve their transparency for the community. Therefore, creating new LLM capability dimensions is not our first priority. Nevertheless, as LLMs are getting more and more powerful, we will continue the efforts of adding new evaluation dimensions in GPT-Fathom to assess advanced LLM capabilities, such as long-context understanding, multi-turn conversation, LLM agent, etc. With these new dimensions, we can study the strength and weakness of leading LLMs more comprehensively.
>
> **Response #2 - Open-sourcing and costs:**
>
> To facilitate the research community for reproducible and systematic LLM evaluation, we have open-sourced GPT-Fathom and will post the link in our paper after the anonymity period. With regard to the effort required to evaluate a new model, our codebase serves as a read-to-use toolkit for evaluating new models. Users just need to implement a completion function, and we provide a step-by-step tutorial in our open-source repo. The evaluation time cost depends on the model size, inference efficiency and available computational resources / API rate limtis.
>
> **Response #3 - Sufficiency of capability dimensions:**
>
> We believe that the chosen capability dimensions cover a broad spectrum of LLM functionalities. We acknowledge that as LLMs rapidly progress, as mentioned above, we will continue the efforts of adding new evaluation dimensions in GPT-Fathom to cover advanced capabilities. Currently, we have already added evaluations of factuality reasoning and long-context understanding, which is available in our repo (link will be posted after the anonymity period).
>
> We hope these responses adequately address your questions. Your feedback is invaluable to us, and we look forward to further discussions on our work.

---

### Official Review · Reviewer_Fohj · 2023-10-28

**Soundness:** 3 good
**Presentation:** 3 good
**Contribution:** 2 fair
**Rating:** 3
**Confidence:** 3

**Summary:**

The paper proposes a new benchmark GPT-Fathom for comparing the performance of closed-source and open-source LLMs. The benchmark is comprehensive and consistently compares different LLMs. The paper has shown the impact of evolution of closed-source LLMs, inclusion of coding datasets during training, as well as alignment methods such as SFT and RLHF.

**Strengths:**

- Creating a well-designed benchmark for LLMs is an important problem statement
- Consideration of both open source and closed source LLMs in the evaluation
- Focus on reproducibility, consistent settings, and ablation of methods/prompts.
- Extensive experiments and analysis of LLMs
- Explanation of why the black box evaluation was considered for all benchmarks and LLMs.
- Interesting analysis of evolution of OpenAI LLMs.

**Weaknesses:**

The research questions addressed by the paper is unclear. For a research publication focused on benchmarking, it is insufficient to study a new set of LLMs and explain the results. The paper needs to explain the benchmark design and how it has resulted in a substantial improvement over existing benchmarks. Listing a few points below.
- The paper claims results are based on "aligned" settings, but still includes numbers from other papers (in brackets) and optimized results (with a star). Instead, it will be useful to compare the numbers in existing papers, and show the impact of the aligned evaluation settings. Did the results change? If so, what was the reason? Such an analysis would confirm their posed hypothesis that aligned evaluation settings lead to more insights than those already published.
- Similarly, it would be good to understand each new feature introduced by GPT-Fathom compared to prior benchmarks, and show why it led to a better evaluation outcome not just for LLMs considered but also for future LLMs that will get evaluated.
- The paper claims that they have picked representative LLMs, benchmarks, and settings. Why are the choices made representative? No explanation have been provided. Without an explanation, the benchmark looks like a collection of benchmarks from other papers, and the benefits of the proposed benchmark is not clear.
- The paper acknowledges that the categories in the benchmarks are chosen based on their "own interpretation". But no justification is provided to explain why this  interpretation should be adopted by the research community as the best benchmark to use for LLMs. Some analysis on how these benchmarks cover the range of tasks that LLMs are used for will be useful.
- The paper repeatedly states that it is trying to answer open questions from the community. The open questions that these benchmarks provide answers for is not clearly stated.
- Prompt sensitivity is an important issue. Two prompts show there is an issue, but it is unclear if LLMs work well for these two prompts that the sensitivity issue is resolved. A better design to evaluate sensitivity with an appropriate metric will be more useful.
- In consistency in the number of shots across benchmarks and types of ablation does not show "aligned" settings claimed by the paper.

**Questions:**

Some clarification questions:
- How are you automating the testing of web-version of LLMs? Is that done manually or through some web toolkit?
- I did not understand what is meant by "pass@K". Do you pick the best answer out of K retries?
- Why does Table 5 and 6 not use zero-shot prompting?
- Why do different benchmarks use different shots?

---

> ### Author Response · Authors · 2023-11-19
> **Official Response to Reviewer Fohj[1/3]**
>
> Thank you for your detailed review and valuable feedbacks. We appreciate the opportunity to clarify and expand on certain aspects of our paper.
>
> **Response #1 - Research questions and improvement over existing benchmarks**
>
> The key research question studied in our work is: How GPT-3 progressively improves to GPT-4? Since OpenAI has not revealed much about their evolutionary path, to answer this question, we systematically evaluate all their major LLMs to break down the impacts of techniques such as adding code data for pretraining, SFT and RLHF. Therefore, instead of just creating yet another LLM leaderboard, the focus of our work is to improve the transparency of leading LLMs (especially closed-source ones) and offer insights for better understanding how their techniques improve LLMs across various capabilities.
>
> Compared to existing works on LLM evaluation, our work improves in the following aspects:
>
> 1. Reproducible evaluation under aligned settings: LLMs are known to be sensitive to the evaluation setting, i.e., different settings and prompts may lead to very different evaluation results. Yet, many existing work simply refer to scores from other papers without consistent settings and prompts, which may easily skew the observations. In contrast, our work provides systematic evaluation of leading LLMs across various capability dimensions, all under aligned settings for fair comparison. We provide an open-source LLM benchmark suite as an off-the-shelf toolkit, so that other researchers can easily reproduce our reported LLM performance, and save a lot of efforts when comparing a new model with existing LLMs.
>
> 2. Novel insights on the impacts of SFT / RLHF: Our work provides in-depth analysis on how SFT / RLHF affects various LLM capabilities, such as SFT having opposite effects on weaker and stronger base models, and RLHF transforming pass@100 (intrinsic coding abilities) into pass@1 (one-take bug-free coding abilities). These novel insights offer valuable guidance for future advancement of LLMs.
>
> 3. Novel challenges of advanced LLMs: Our work discovers novel challenges that leading models have to address, such as the "seesaw phenomenon" in LLM capabilities, where certain capabilities improve, while a few other capabilities clearly regress. We demonstrate the existence of this phenomenon in both leading closed-source and open-source LLMs, inspiring the research community to dedicate more efforts to tackle this challenge.
>
> 4. Novel study on model sensitivity: Previous works on LLM evaluation rarely study the sensitivity of model, such as sampling variance and prompt sensitivity. In fact, many existing LLM leaderboards reference scores from other papers without consistent settings and prompts, which may inadvertently encourage cherry-picking favored settings and prompts for better results. In contrast, we demonstrate that even leading open-source LLMs are susceptible to prompt variations: a minor alteration of the prompt could cause the LLM to fail in solving the problem. We highlight the model sensitivity issue with our experimental results, and urge the community to improve the robustness of LLMs.
>
> To summarize, GPT-Fathom is not just a collection of benchmarking results. We offer novel insights for community-concerned questions and discover novel challenges, aiming to improve the transparency of leading LLMs and inspire further advancements of LLMs.
>
> We hope the clarifications above address your concerns about the research questions and novelty of our work. We appreciate your helpful feedbacks in refining our submission.

---

> ### Author Response · Authors · 2023-11-19
> **Official Response to Reviewer Fohj[2/3]**
>
> **Response #2 - Numbers from other papers**
>
> We apologize for the confusion about the evaluation settings. We clarify our choice as follows.
>
> We refer to numbers from PaLM 2-L and Claude 2 papers since we currently have no access to their APIs (Google only provides access to smaller PaLM 2 models; Claude 2 only provides API access to organization users). We try our best to align our evaluation settings with these two models, such as the number of "shots", whether CoT prompting is used, and the evaluation metrics. Since PaLM 2-L and Claude 2 have different evaluation settings in their official papers, we separate them into two Tables (please refer to Table 1 and Table 2) and in each table, we evaluate other models with aligned settings on every benchmark for fair comparison.
>
> For our own reported results, we strictly align all the evaluation settings and prompts, with the only exceptions for Llama models, where a few numbers are marked with * where we use optimized prompts. This is because we find that Llama models are quite prompt-sensitive, while OpenAI's models are pretty robust. If we randomly pick a prompt, the numbers of Llama models could be significantly lower than their officially reported scores, which may incur challenges of fairness of our evaluation suite. In fact, we study the model sensitivity in detail in later part of Sec 3.2 (please refer to Table 6), and the original scores without using optimized prompts are listed there.
>
> A direct comparison with numbers from other papers is challenging due to the lack of released prompts and few-shot samples in these papers. This lack of transparency in benchmark settings underlines the necessity of our open-source reproducible LLM benchmark suite. Nevertheless, we do carefully compare our evaluation results with officially reported numbers, and most our results match the reported performance in official papers, within a margin of slight deviation.
>
> **Response #3 - Selection criteria for benchmarks and LLMs**
>
> At the beginning of Sec 2.2, we describe our criteria for benchmark selection: 1) cover as many aspects of LLM capabilities as possible; 2) adopt widely used benchmarks for LLM evaluation; 3) clearly distinguish strong LLMs from weaker ones; 4) align well with the actual usage experience of LLMs. Following these criteria, we carefully pick benchmarks in our evaluation suite. We admit that there is no "perfect" selection of benchmarks for any evaluation framework, since both benchmarks and LLMs are evolving. We are currently working on latest benchmarks such as SciBench, SummEdits and QuALITY for evaluating advanced reasoning and long-context capabilities of LLMs.
>
> With regard to the selection criteria of LLMs, we also have a brief discussion in Sec 2.1 that our goal is to help the community better understand how OpenAI improves from GPT-3 to GPT-4 and pinpoint the position of LLMs under development. To achieve this goal, we mainly consider evaluating these types of LLMs: 1) OpenAI’s leading models, such as GPT-4; 2) OpenAI’s major earlier models, such as GPT-3 and InstructGPT; 3) other leading closed-source models, such as PaLM 2 and Claude 2; 4) leading open-source models, such as LLaMA and Llama 2. Since LLMs are still rapidly evolving, we continue the effort of evaluating cutting-edge LLMs, such as GPT-4-Turbo from OpenAI. In our updated submission, we have added the evaluation results of gpt-3.5-turbo-instruct-0914, a new model from OpenAI which was just released in 2023/10.
>
> **Response #4 - Benchmark categories**
>
> For the categories of benchmarks, we investigate many existing works especially the papers of OpenAI models, Claude 2, PaLM 2, LLaMA and Llama 2, which agree on a reasonable categorization of popular benchmarks. For example, BBH is often attached to the "reasoning" category, while GSM8K is clearly associated with the "math" category. We do admit that our categorization of benchmarks is by no means the exclusive approach, but a reasonable and commonly acceptable one.
>
> **Response #5 - Addressing open questions**
>
> We apologize for the confusion. The key question that the community is currently eager to know is how GPT-3 progressively improves to GPT-4, including technical details such as whether adding code data improves LLM's reasoning capability, which aspects of LLM capability can be improved by SFT and RLHF, how much is the alignment tax, etc. We briefly introduce these open questions in the introduction, and detail them in Section 3.2.
>
> **Response #6 - Prompt sensitivity**
>
> Your point on prompt sensitivity is well-taken. We acknowledge this as a valuable area for future research. Due to resource constraints, we could not comprehensively benchmark the prompt sensitivity of all models. However, we do clearly demonstrate the existence of this issue, and inspire the research community to address this challenge in the future.

---

> ### Author Response · Authors · 2023-11-19
> **Official Response to Reviewer Fohj[3/3]**
>
> **Response #7 - Aligned settings and ablation studies**
>
> Our "aligned setting" refers to the main evaluation results of LLMs on each benchmark are obtained by the same prompt, same number of "shots", same sampling hyperparameter, and same metrics, etc. Please note that these evaluation settings are aligned within each benchmark, but may differ across different benchmarks. For example, we adopt 1-shot setting for most benchmarks, but follow the most commonly used 5-shot setting for MMLU. In the ablation studies, we do alter the number of "shots" since we need to investigate how it impacts the performance of LLMs.
>
> **Response #8 - Clarification questions**
>
> Automating Web-version testing: We wrapped up the Web-version ChatGPT (including GPT-3.5 and GPT-4) into a local API for the ease of bulk evaluation, following some open-source projects such as ninja (https://github.com/gngpp/ninja).
>
> Clarification on "pass@k" metric: This is a widely used metric in evaluating the coding capability of LLMs since [1], and we follow the standard definition of pass@k in our work: for each coding problem, k code samples are generated, and a problem is considered solved if any sample passes the unit tests, and the total fraction of problems solved is reported. We have added the definition of pass@k in the "Coding" paragraph of Sec 2.2 in our updated submission. Thank you for your feedback helping us improve the clarity of our work.
>
> Number of "shots" in Tables 5 & 6: We report the results of our ablation study on how CoT prompting and prompt variation impacts LLM performance in Table 5 & 6, respectively, and we keep the number of "shots" consistent with our main setting (as in Table 1) rather than zero-shot. We do have the zero-shot setting in another ablation study on the impact of number of "shots" (Table 4), where we report the zero-shot performance on ARC-c and HellaSwag.
>
> Different benchmarks use different "shots": Our ablation study on the number of "shots" (Table 4) shows that the performance of LLMs struggles to improve beyond 1-shot, particularly for stronger models. This indicates that 1-shot typically works well, which is our primary evaluation setting for most benchmarks (Table 1). However, for some benchmarks, there exists widely adopted number of "shots" in existing works, such as 5-shot for MMLU, 8-shot CoT for GSM8K, 0-shot for HumanEval, etc., and we follow these widely adopted "shots" for ease of comparing our results to existing reported scores. Note that these number of "shots" are also adopted in the paper of PaLM 2, so that we can compare with its performance using the same number of "shots" in Table 1.
>
>
> We hope our responses above adequately address your concerns and clarify the aspects you highlighted. Your feedback has been instrumental in refining our work and we have made the suggested modifications based on your valuable feedback.
>
>
> [1] Chen, Mark, et al. “Evaluating Large Language Models Trained on Code.” arXiv, 7 July 2021, http://arxiv.org/abs/2107.03374. arXiv.

---

> ### Author Response · Authors · 2023-11-22
>
> Dear Reviewer Fohj, we revised our paper with the comparison of our scores and the official scores (refer to the *General Response to Reviewers*). Please let us know your thoughts, and we are more than happy to answer any further questions.

---

> ### Comment · Reviewer_Fohj · 2023-11-22
> **Thank you for your response**
>
> Responding to each aspect discussed in your comment.
> - "How GPT-3 progressively improves to GPT-4?" is not a research question. "Does adding code data to pretraining improve performance"? is a research question. The research question needs to specific, measurable with a metric, and outcome of the result should be clear. So you are actually addressing three research questions here. What is not clear to me is: have these not been studied before?
> 1. For the first comment, I was expecting a table with model name as columns and prior benchmark papers as rows, and finally your benchmark results at the bottom. I would like to see the impact of the "aligned settings" on the results compared to prior works. Your contribution hinges on a better benchmarking setup, but I'm not able to discern how much it is better compared to prior benchmarking setups where the prompts, shots, etc. were not aligned.
> 2. Can you make conclusions with statistical tests? The hypothesis "RLHF improves LLM capabilities" should have a yes or no answer based on the result of this test.
> 3. Are you the first to discover the "seesaw phenomenon"? Can you confirm this seesaw phenomenon consistently occurs with a statistical test?
> 4. Just saying the results change with change with prompt is not enough. What is your metric to measure sensitivity? How much sensitivity is too much to declare that LLMs are sensitive?

---

> > ### Author Response · Authors · 2023-11-22
> >
> > Thank you for reading our responses and follow up with further discussions. We respond to your comments as follows.
> >
> > * These questions have definitely been studied before as individual research topics. However, by "how GPT-3 progressively improves to GPT-4", we mean that what we really would like to study is actually how these well-known techniques combine and contribute to the success of the remarkably powerful model GPT-4. This "how" question is not directly researchable due to the lack of transparency of OpenAI, so we break it down into several sub-questions like "whether adding code into pretraining improves performance" and "if yes, how much improvement", etc. One way of investigating these sub-questions is to conduct experiments of our own and analyze the performance gain, but the methodology used could be very different from OpenAI's, leading to completely wrong conclusions. Another way is to retrospectively benchmark the performance of all OpenAI's major models, and shed some light on this mysterious "evolutionary path". To the best of our knowledge, our work is an initial study following the second research trail.
> >
> > Other questions:
> >
> > 1. "Aligned setting" should be a basic yet critical requirement for any LLM evaluation works to ensure an apple-to-apple comparison. For example, the performance on MMLU under 0-shot setting may not be compared to that under 5-shot setting. Our key contribution is not to demonstrate the benefit of "aligned setting", which is purely logical. Instead, we provide a fully reproducible evaluation toolkit that our settings are not only internally aligned, but also verified to produce results that can externally match the official scores of leading LLMs, such as GPT-4.
> >
> >     To the best of our knowledge, we are the first LLM evaluation work that simultaneously achieve the followings: 1) fully open-source and reproducible; 2) evaluation settings are aligned on every benchmark; 3) evaluation results match official scores of leading LLMs; 4) cover all major OpenAI models from GPT-3 to GPT-4 for retrospective study. These important desiderata show that our work significantly improves over prior LLM leaderboards. As for "a similar table comparing with other benchmarks", it is infeasible to make such comparisons since other benchmarks have not covered all the desiderata listed above.
> >
> > 2. We only evaluated 20+ benchmarks, which may not support a solid hypothesis test. Also, even we do some statistical test and come to a plausible conclusion, e.g., "RLHF can improve LLMs overall", this may not be helpful at all to the community. What really matters is to break down the impacts of RLHF on every dimension of LLM capabilities, such as our analysis on how RLHF transfers pass@100 (as intrinsic coding skill) to pass@1 (as one-pass bug-free capability). With these detailed analysis, the community knows way more than just a "yes or no" conclusion from some statistical test. Readers of our work may correctly apply RLHF to relevant tasks where it can indeed boost the performance.
> >
> > 3. To the best of our knowledge, our work is the first to discover that even GPT-4 has the "seesaw phenomenon" based on our systematic evaluations. We only have access to two GPT-4 API versions, `gpt-4-0314` and `gpt-4-0613`, which may not support a solid statistical test of whether this phenomenon consistently occurs. Since OpenAI officially admits this issue (please refer to https://openai.com/blog/function-calling-and-other-api-updates), we believe that the seesaw phenomenon is not a rare problem.
> >
> > 4. It may be very challenging to "prove" that one LLM is not prompt-sensitive, however, just one counter example (e.g., our case study in Table 6, a slight alteration of the prompt template results in the score of Llama 2-70B on TriviaQA plummeting from 74.0 to 55.5) is enough to demonstrate that one LLM is indeed prompt-sensitive. As for the quantitative metric to assess the level of prompt sensitivity, this could be a great research topic but out of the scope of our work.

---

### Official Review · Reviewer_aiZP · 2023-10-31

**Soundness:** 2 fair
**Presentation:** 3 good
**Contribution:** 2 fair
**Rating:** 3
**Confidence:** 4

**Summary:**

This paper curates a benchmark suite to evaluate the performance of LLMs.

**Strengths:**

The proposed benchmark covers a range of aspects to study, including knowledge, math, coding, etc.

It also provides performance and analysis of several popular LLMs on the proposed benchmark.

**Weaknesses:**

I appreciate the experiments and analysis, but I am mostly concerned with mismatched claims and unclear novelty.

1. mismatched claimed: The paper underscores that the paper sheds light on "the evolutionary path from GPT-3 to GPT-4," several times in the abstract, intro, and conclusion. However, after reading the main text, I could not find enough evidence and/or analysis on the evolutionary path. Figure 1 gives a visualization of OPENAI's announcements of different features/models over time, which the authors defined as evolutionary path. But how is it related to the proposed benchmark?

2. unclear novelty: The proposed benchmark, GPT-Fathom, is effectively a selection/collection of (subsets of) existing benchmark datasets (MMLU, Bigbench, etc). Prompting and evaluation metrics are also quite standard. The analysis seems to resonate many well-known assertions, e.g., proprietary models are more performant. It is unclear to me what new message this paper brings in.

**Questions:**

Please see my question in the weakness parts.

---

> ### Author Response · Authors · 2023-11-19
> **Official Response to Reviewer aiZP**
>
> Thank you for your insightful review and feedbacks. We value the effort and time you dedicated to evaluating our submission and have carefully considered your comments. Here are our responses to the main issues raised.
>
> **Response #1 - Evolutionary path**
>
> We apologize for the confusion of the "evolutionary path from GPT-3 to GPT-4." To clarify, Figure 1 is not just a timeline of OpenAI's feature announcements. We use Figure 1 to visualize the major OpenAI models that are evaluated in our paper, and highlight the key techniques developed along the path from GPT-3 to GPT-4, e.g., adding code data for pretraining, using SFT / RLHF for fine-tuning. It is well-known that GPT-4 dramatically outperforms GPT-3, however, OpenAI has not revealed much about how they get there step-by-step. We evaluated all the major models between GPT-3 and GPT-4 to decipher their "evolutionary path", i.e., how each technique contributes to the advancement from GPT-3 to GPT-4. By providing a systematic evaluation suite, we aim to shed lights on the impacts of the key techniques used by advanced LLMs and improve the transparency of closed-source models.
>
> **Response #2 - Novelty**
>
> 1. Beyond well-known facts: LLMs are known to be sensitive to the evaluation setting, i.e., different settings and prompts may lead to very different evaluation results. Yet, many existing work simply refer to scores from other papers without consistent settings and prompts, which may easily skew the observations. In contrast, our work provides systematic evaluation of leading LLMs across various capability dimensions, all under aligned settings for fair comparison.
>
>     Our work is not just resonating some well-known facts like proprietary models perform better. Instead, we provide an open-source LLM benchmark suite as an off-the-shelf toolkit, so that other researchers can not only easily reproduce our reported LLM performance, but also save a lot of efforts when they need to compare a new model with existing LLMs across various capabilities. To cover the common use cases, we intend to select popular benchmarks with standardized settings and widely adopted metrics.
>
> 2. Novel insights on the impacts of SFT / RLHF: Our work provides in-depth analysis on how SFT / RLHF affects various LLM capabilities, such as SFT having opposite effects on weaker and stronger base models, and RLHF transforming pass@100 (intrinsic coding abilities) into pass@1 (one-take bug-free coding abilities). These novel insights offer valuable guidance for future advancement of LLMs.
>
> 3. Novel challenges of advanced LLMs: Our work discovers novel challenges that leading models have to address, such as the "seesaw phenomenon" in LLM capabilities, where certain capabilities improve, while a few other capabilities clearly regress. We demonstrate the existence of this phenomenon in both leading closed-source and open-source LLMs, inspiring the research community to dedicate more efforts to tackle this challenge.
>
> 4. Novel study on model sensitivity: Previous works on LLM evaluation rarely study the sensitivity of model, such as sampling variance and prompt sensitivity. In fact, many existing LLM leaderboards reference scores from other papers without consistent settings and prompts, which may inadvertently encourage cherry-picking favored settings and prompts for better results. In contrast, we demonstrate that even leading open-source LLMs are susceptible to prompt variations: a minor alteration of the prompt could cause the LLM to fail in solving the problem. We highlight the model sensitivity issue with our experimental results, and urge the community to improve the robustness of LLMs.
>
>     To summarize, GPT-Fathom is not just a collection of benchmarking results. We build open-source toolkit for reproducible LLM evaluation, offer novel insights for community-concerned questions, and discover novel challenges, aiming to improve the transparency of leading LLMs and inspire further advancements of LLMs.
>
>
> We hope our responses above address your concerns and demonstrate the novelty of our work. We appreciate your helpful feedbacks in refining our submission.

---

### Author Response · Authors · 2023-11-22
**General Response to Reviewers[1/2]**

We thank all reviewers for the constructive comments. We have revised our paper, adding Appendix E and F for **our evaluation of Llama/LLaMA 2 Family**, and **the comparison between our scores and the official scores**.

**Evaluation of Llama/LLaMA 2 Family**
We evaluate the entire LLaMA / Llama 2 family, including models ranging from 7B to 65B / 70B parameters, and report the complete results in the following table.

| Capability Category |  | Benchmark | Setting | LLaMA-7B | Llama 2-7B | LLaMA-13B | Llama2-13B | LLaMA-30B | LLaMA-65B | Llama 2-70B |
|:---:|:---:|:---:|:---:|:---:|:---:|:---:|:---:|:---:|:---:|:---:|
| Knowledge | Question Answering | Natural Questions | 1-shot | 17.6 | 19.8 | 20.8 | 27.6 | 24.0 | 27.7 | 27.0 |
|  |  | WebQuestions | 1-shot | 37.0 | 38.3 | 37.6 | 42.8 | 39.0 | 42.2 | 38.2 |
|  |  | TriviaQA | 1-shot | 52.0 | 61.1 | 66.6 | 70.0 | 73.5 | 73.4 | 74.0 |
|  | Multi-subject Test | MMLU | 5-shot | 25.1 | 41.0 | 38.5 | 49.5 | 51.0 | 60.1 | 67.8 |
|  |  | AGIEval-EN | few-shot | 19.1 | 25.7 | 27.0 | 35.7 | 34.7 | 38.0 | 44.0 |
|  |  | ARC-e | 1-shot | 30.0 | 62.3 | 67.6 | 76.4 | 82.4 | 87.2 | 93.4 |
|  |  | ARC-c | 1-shot | 26.7 | 48.6 | 49.1 | 55.7 | 60.8 | 71.8 | 79.6 |
| Reasoning | Commonsense Reasoning | LAMBADA | 1-shot | 19.0 | 38.0 | 47.0 | 56.4 | 32.5 | 30.9 | 30.4 |
|  |  | HellaSwag | 1-shot | 24.6 | 25.4 | 28.9 | 37.2 | 31.3 | 47.8 | 68.4 |
|  |  | WinoGrande | 1-shot | 50.4 | 50.2 | 48.1 | 52.1 | 51.3 | 54.6 | 69.8 |
|  | Comprehensive Reasoning | BBH | 3-shot CoT | 33.7 | 38.4 | 39.1 | 46.2 | 49.6 | 58.2 | 65.0 |
| Comprehension | Reading Comprehension | RACE-m | 1-shot | 26.7 | 45.8 | 52.4 | 57.9 | 65.3 | 77.0 | 87.6 |
|  |  | RACE-h | 1-shot | 29.1 | 39.5 | 48.5 | 55.1 | 64.1 | 73.0 | 85.1 |
|  |  | DROP | 3-shot, F1 | 9.6 | 7.7 | 8.7 | 9.3 | 9.8 | 56.4 | 67.6 |
| Math | Mathematical Reasoning | GSM8K | 8-shot CoT | 13.9 | 17.2 | 18.4 | 28.6 | 35.1 | 53.6 | 56.4 |
|  |  | MATH | 4-shot CoT | 0.4 | 0.1 | 0.4 | 0.5 | 0.5 | 2.6 | 3.7 |
| Coding | Coding Problems | HumanEval | 0-shot, pass@1 | 7.0 | 14.6 | 9.7 | 15.8 | 7.2 | 10.7 | 12.7 |
|  |  | MBPP | 3-shot,pass@1 | 23.7 | 39.2 | 29.5 | 46.0 | 38.5 | 44.8 | 58.0 |
| Multilingual | Multi-subject Test | AGIEval-ZH | few-shot | 22.3 | 23.4 | 23.5 | 29.7 | 28.4 | 31.7 | 37.9 |
|  |  | C-Eval | 5-shot | 11.5 | 10.3 | 14.8 | 28.9 | 10.1 | 10.7 | 38.0 |
|  | Mathematical Reasoning | MGSM | 8-shot CoT | 2.7 | 2.3 | 2.8 | 4.1 | 3.1 | 3.6 | 4.0 |
|  | Question Answering | TyDi QA | 1-shot, F1 | 2.4 | 3.6 | 3.2 | 4.5 | 3.8 | 12.1 | 18.8 |
| Safety | Truthfulness | TruthfulQA | 1-shot | 37.6 | 31.0 | 29.5 | 38.0 | 44.5 | 51.0 | 59.4 |
|  | Toxicity | RealToxicityPrompts | 0 -shot | 14.5 | 14.8 | 14.9 | 14.8 | 14.7 | 14.8 | 15.0 |

---

### Author Response · Authors · 2023-11-22
**General Response to Reviewers[2/2]**

**Our Scores vs. Official Scores**

To verify the correctness of our implementation, we compare our evaluation results with the officially reported scores from GPT-4 technical report and Microsoft's early experiments with GPT-4. To ensure an apple-to-apple comparison, we align the evaluation settings on each benchmark, as summarized in the following table. This head-to-head comparison demonstrates that our evaluation results are consistent with the official scores, within a margin of slight deviation. Since the official prompts and in-context examples for evaluation are not publicly available, the slight deviation is totally reasonable. We also notice that the performance gain with in-context examples beyond 1-shot is pretty marginal, which aligns with our primary evaluation setting.

| Benchmark | Setting    | gpt-4-0314 (our evaluation) | GPT-4 (official score) |
|-----------|------------|----------------------------|-----------------------|
| MMLU      | 5-shot     | 83.7                       | 86.4                  |
| ARC-c     | 25-shot    | 96.3                       | 95.6                  |
|           | 1-shot     | 94.9                       | -                     |
| HellaSwag | 10-shot    | 92.5                       | 95.3                  |
|           | 5-shot     | 92.4                       | -                     |
| DROP      | 1-shot     | 89.3                       | 87.5                  |
| GSM8K     | 3-shot, F1 | 86.7                       | -                     |
|           | 8-shot CoT | 78.7                       | 80.9                  |
| HumanEval | 4-shot CoT | 91.6                       | 92.0                  |

We also compare our evaluation results with the official scores reported in LLaMA and Llama 2. Similarly, in the table below, we report the benchmarks whose official evaluation settings match our settings, and compare our results with the official scores. We observe that on some benchmarks, such as BBH, our results are higher than the official scores; while on some other benchmarks, such as TriviaQA and MATH, our results are lower than the official scores. This phenomenon is consistent with our conclusion that LLaMA and Llama 2 are pretty prompt-sensitive. To be more specific, take MATH as an example, since we use the exact same setting and prompt as we evaluate OpenAI models on this benchmark, and our evaluation result of GPT-4 matches the official scores, we argue that the prompt sensitivity of LLaMA / Llama 2 models explains the performance gap of our evaluation and their official scores.

For coding benchmarks HumanEval and MBPP, the official LLaMA and Llama 2 papers use different temperature $T$ to evaluate pass@1 ($T=0.1$) and pass@100 ($T=0.8$). In contrast, we follow OpenAI's setting on coding evaluation and uniformly use $T=0.8$ for all our evaluations on coding benchmarks. This explains the performance difference of our results and the official scores of LLaMA and Llama 2 on HumanEval and MBPP.


| Benchmark | Setting | LLaMA-65B (our evaluation) | LLaMA-65B (official score) | Llama 2-70B (our evaluation) | Llama 2-70B (official score) |
| :---: | :---: | :---: | :---: | :---: | :---: |
| Natural Questions | 1-shot | 27.7 | 31.0 | 27.0 | 33.0 |
| TriviaQA | 1-shot | 73.4 | 84.5 | 74.0 | 85.0 |
| MMLU | 5-shot | 60.1 | 63.4 | 67.8 | 68.9 |
| BBH | 3-shot CoT | 58.2 | 43.5 | 65.0 | 51.2 |
| GSM8K | 8-shot CoT | 53.6 | 50.9 | 56.4 | 56.8 |
| MATH | 4-shot CoT | 2.6 | 10.6 | 3.7 | 13.5 |
| HumanEval | 0-shot,pass@1 | 10.7 $(T=0.8)$ | 23.7 $(T=0.1)$ | 12.7 $(T=0.8)$ | 29.9 $(T=0.1)$ |
| MBPP | 3-shot, pass@1 | 44.8 $(T=0.8)$ | 37.7 $(T=0.1)$ | 58.0 $(T=0.8)$ | 45.0 $(T=0.1)$ |

We believe we have addressed each reviewer's concerns and questions in our individual responses. Please let us know if you have any other questions. We look forward to discussing with the reviewers to make the paper better.

---

> ### Comment · Reviewer_Fohj · 2023-11-22
> **Thanks for the comparative analysis**
>
> I was looking for something like this in the paper. So this is a good addition, it shows you can replicate results. Some suggestions below:
> 1. Instead of conclusions such as "slight deviation" and "pretty marginal", make conclusions with statistical tests.
> 2. Is the second table, are you following the same prompts as the official benchmark?
> 3. Experiments here are not sufficient to conclude prompt sensitivity, that requires a different experiment with its own hypothesis testing.
> 4. I was looking for a similar table comparing your benchmark with the other open source benchmarks. This would demonstrate that your aligned setting did improve on prior benchmarks. I still do not see conclusive evidence that you are improving on other benchmarks.

---

> ### Author Response · Authors · 2023-11-22
> **Responses**
>
> Thank you for your insightful suggestions. We respond to your comments one-by-one as follows.
>
> 1. We never treat official score as "ground truth", i.e., deviation from that does not mean anything like "error" or "variance". As we claimed above, the official prompts and in-context examples are not released. For most LLMs, different prompts lead to different results, and we don't think statistical test is helpful to compare two results that are supposed to be different already. The really big issue here is the non-transparency and non-reproducible evaluation results. There could be the risk of cherry picking / prompt engineering for better results. A key contribution of our work is the fully open-source and reproducible evaluation toolkit. We just use the official score as a reference for "sanity check", i.e., make sure our implementation produces reasonable results. Although this sounds easy, existing works on LLM evaluation may not pass this line.
>
>     For instance, the latest HELM leaderboard v4.0 (https://crfm.stanford.edu/helm/latest/#/leaderboard), which was just updated days ago, reports that Llama 2-70B achieves 58.2 on MMLU, while the official score is 68.9 and our result is 67.8 as in the second table above. Let's say someone has a new LLM and would like to compare with existing leading LLMs. If this new LLM achieves 62.0 evaluated by HELM v4.0, can we say it significantly beats Llama 2-70B? Probably not, since Llama 2-70B can actually achieve even higher score than 62.0 not only from the official paper but also reproduced in GPT-Fathom. This highlights our contributions on reproducing the evaluation results, and then make our results fully open-source and reproducible.
>
> 2. As mentioned above, the prompts used in official scores are not released, and most benchmark datasets have not provided an official prompt for evaluation. We do follow the official prompt if the benchmark happens to provide it, as described in Table 8 of our paper.
>
> 3. The tables here are just to compare our results vs. official scores, and are not trying to conclude the prompt sensitivity of any LLMs. In fact, we experimentally study the prompt sensitivity in Table 6. Note that it may be very challenging to "prove" that one LLM is not prompt-sensitive, however, just one counter example (e.g., our case study in Table 6, a slight alteration of the prompt template results in the score of Llama 2-70B on TriviaQA plummeting from 74.0 to 55.5) is enough to demonstrate that one LLM is indeed prompt-sensitive. As for the quantitative metric to assess the level of prompt sensitivity, this could be a great research topic but out of the scope of our work.
>
> 4. "Aligned setting" should be a basic yet critical requirement for any LLM evaluation works to ensure an apple-to-apple comparison. For example, the performance on MMLU under 0-shot setting may not be compared to that under 5-shot setting. Our key contribution is not to demonstrate the benefit of "aligned setting", which is purely logical. Instead, we provide a fully reproducible evaluation toolkit that our settings are not only internally aligned, but also verified to produce results that can externally match the official scores of leading LLMs, such as GPT-4.
>
>     To the best of our knowledge, we are the first LLM evaluation work that simultaneously achieve the followings: 1) fully open-source and reproducible; 2) evaluation settings are aligned on every benchmark; 3) evaluation results match official scores of leading LLMs; 4) cover all major OpenAI models from GPT-3 to GPT-4 for retrospective study. These important desiderata show that our work significantly improves over prior LLM leaderboards. As for "a similar table comparing with other benchmarks", it is infeasible to make such comparisons since other benchmarks have not covered all the desiderata listed above.